# Machine Learning Generation of Dynamic Protein Conformational Ensembles

**DOI:** 10.3390/molecules28104047

**Published:** 2023-05-12

**Authors:** Li-E Zheng, Shrishti Barethiya, Erik Nordquist, Jianhan Chen

**Affiliations:** 1Department of Gynecology, The First Affiliated Hospital of Fujian Medical University, Fuzhou 350005, China; zelie@sina.com; 2Department of Chemistry, University of Massachusetts Amherst, Amherst, MA 01003, USA; sbarethiya@umass.edu (S.B.); enordquist@umass.edu (E.N.)

**Keywords:** autoencoder, Boltzmann generator, collective variable, dimension reduction, enhanced sampling, generative adversarial network, latent space, neural network, physics-informed machine learning, transfer learning

## Abstract

Machine learning has achieved remarkable success across a broad range of scientific and engineering disciplines, particularly its use for predicting native protein structures from sequence information alone. However, biomolecules are inherently dynamic, and there is a pressing need for accurate predictions of dynamic structural ensembles across multiple functional levels. These problems range from the relatively well-defined task of predicting conformational dynamics around the native state of a protein, which traditional molecular dynamics (MD) simulations are particularly adept at handling, to generating large-scale conformational transitions connecting distinct functional states of structured proteins or numerous marginally stable states within the dynamic ensembles of intrinsically disordered proteins. Machine learning has been increasingly applied to learn low-dimensional representations of protein conformational spaces, which can then be used to drive additional MD sampling or directly generate novel conformations. These methods promise to greatly reduce the computational cost of generating dynamic protein ensembles, compared to traditional MD simulations. In this review, we examine recent progress in machine learning approaches towards generative modeling of dynamic protein ensembles and emphasize the crucial importance of integrating advances in machine learning, structural data, and physical principles to achieve these ambitious goals.

## 1. Introduction

Proteins are the major functional macromolecules in biology, which play critical and diverse roles in virtually all cellular processes and are involved in numerous human diseases, including cancers, neurodegenerative diseases, and diabetes [1,2,3]. A central property of proteins is that their amino acid sequence (and thus their chemical structure) encodes highly specific three-dimensional (3D) structural properties to support their function. Enormous efforts have been invested in experimental determination of the high-resolution structures of proteins, using a range of techniques, including nuclear magnetic resonance (NMR), X-ray crystallography, and more recently, cryogenic electron microscopy (Cryo-EM) [4,5]. These efforts have now provided an arguably complete coverage of all protein families and possible folds, with over 200,000 protein structures publicly available through the RCSB Protein Data Bank (PDB) database [6]. In parallel with these developments, dramatic advances have been made in leveraging available structures and multi-sequence alignments for the prediction of protein structure from sequence information alone [7,8]. These efforts culminated in recent development of AlphaFold [9] and RoseTTAFold [10], which are end-to-end deep machine learning (ML) methods capable of generating high-quality structures for the entire proteomes [11]. Most recently, large language models have also emerged as powerful ML tools for discovering structural and functional properties of proteins from massive sequence databases [12]. For example, ESMfold from Meta trained with a masked language modeling objective can develop attention patterns that capture structure contacts and recover atomic protein structures that are comparable to AlphaFold2 predictions [13]. Together, these powerful tools have drastically expanded the structural coverage of proteins [6,14] and are having transformative impacts in biological and biomedical research [15,16].

Notwithstanding the remarkable successes of single protein structure prediction [17], the need for additional developments is well-recognized [18,19,20,21]. In particular, existing structure prediction tools largely aim to generate a single structure for a given sequence; yet, there is not a single “native” state for all proteins [22]. The structures of proteins can change, depending on the environment, such as changes in temperature, pH, or ligand binding, as well as post-translational modifications (PTMs) [23]. More fundamentally, proteins are dynamic in nature and their dynamic properties are essential to how proteins work in biology and how they can be targeted for therapeutic interventions [24]. NMR relaxation analysis is one of the most powerful approaches for deriving the magnitude and timescale of internal protein motions at residue level [25,26,27]. Multiple structures can be determined for various functional states of the same protein. Nonetheless, experimental characterization of dynamic properties and conformational transitions of proteins is challenging and severely limited in spatial and temporal resolutions [28]. Instead, physics-based molecular modeling and simulation have been the workhorses for generating ensembles of dynamic structures and conformational transition paths of proteins at atomistic resolutions [29,30,31,32,33]. These simulations have greatly benefited from efficient GPU-accelerated molecular dynamics (MD) algorithms [34,35,36,37,38,39], advanced sampling techniques [40,41,42,43,44,45,46,47], and steadily improved general-purpose protein force fields [48,49,50]. The reach of MD simulations has also been drastically expanded by the development of the special-purpose Anton supercomputers [51]. Despite these advances, a persisting bottleneck of atomistic MD simulations for generation of dynamic protein ensembles is the computational cost. In general, comprehensive sampling of the dynamic conformational ensemble is only feasible for small and simple systems. As such, there has been a long history and great need of leveraging data-driven ML methods to accelerate MD simulations and/or to directly generate dynamic protein ensembles [52,53,54,55,56,57].

In this short review, we summarize recent progresses in the development and application of deep generative models for biomolecular modeling, particularly those developed for generating dynamic structure ensembles of proteins in various contexts. We note that several outstanding reviews already provide in-depth discussions of how ML can be used to learn the energy landscape and collective variables (CVs), derive coarse-grained models, and generate protein structure ensembles [52,53,54,55,56,57,58,59]. This review will mainly focus on the most recent works published in the last three years, as identified from Web of Science searches using various combinations of keywords, including “machine learning”, “protein conformation”, “protein structure”, and “ensemble” as well as references of related papers. We will also discuss the challenges of generative models for complex protein ensembles and how the incorporation of physical knowledge may be critical for overcoming these limitations.

## 2. A Rich Continuum of Protein Structures and Dynamics for Function

As illustrated in Figure 1, the range of protein conformational dynamics in nature can be roughly classified into four general categories of increasing complexity and thus difficulty for characterization and prediction. The simplest case is local conformational dynamics within a largely well-defined native fold. Such dynamics include atomic thermal fluctuations around the native structure, which measure the local rigidity. Such rigidity information can often be inferred from the crystal B-factors [60] or derived readily from short MD simulations. More importantly, certain local regions, such as loops of a protein, can have nontrivial dynamic properties and sample a range of conformations relevant to the function. For example, the anti-apoptotic Bcl-xL protein [61] contains a BH3-only protein binding interface that adopts many different conformations within the ~50 experimental structures in PDB (Figure 1A). Atomistic simulations with enhanced sampling show that this interface is inherently dynamic and suggests many rapidly interconverting conformations [62,63]. Interestingly, all previous observed conformers are well-represented in the MD-generated ensemble, highlighting the importance of predicting and generating dynamic ensembles of local loops or regions for understanding protein function. Note that simulation of the dynamic ensemble for even a relatively modest local region is computationally intensive, requiring over 16 μs sampling time in the case of Bcl-xl, even with enhanced sampling [63].

The second major class of functional dynamics include proteins that undergo large-scale conformational transitions between two or more major states, which can be triggered by a wide range of cellular stimuli, including ligand binding, PTMs, and changes in the solution conditions (e.g., pH, temperature, and ionic strength) [64,65,66]. Figure 1B illustrates a drastic conformational transition of the COVID-19 spike protein trimer in the pre- and post-fusion states, as driven by interaction with the host membrane [67]. Understanding the molecular mechanisms and details of these large-scale conformational transitions is crucial for understanding protein function and for developing rational strategies of therapeutic interventions targeting these proteins. Experimentally, it may be possible to capture different conformations that correspond to various function states, but some states may require conditions difficult to replicate under structural determination conditions and these states may only be transiently accessible [65,68]. It is even more challenging to experimentally resolve the transition pathways [69,70] and molecular modeling, and simulations are generally required [71,72]. As will be discussed further, this has been one of the areas in which ML and generative models have made major impacts, especially when combined with MD simulations [52,53,58].

The third and fourth classes of functional protein dynamics include proteins that can remain partially or fully disordered under physiological conditions [73,74,75]. These proteins are referred to as intrinsically disordered proteins (IDPs) and are the most challenging to characterize, both experimentally and computationally. These proteins make up ~30% of all eukaryotic proteins and are key components of the regulatory networks that dictate virtually all aspects of cellular decision-making [76]. Deregulated IDPs are associated with many diseases including cancers, diabetes, and neurodegenerative and heart diseases [77,78,79]. Importantly, as illustrated in Figure 1C, IDPs must be described using dynamic structural ensembles. These ensembles are not random and often contain nontrivial transient local and long-range structures that are crucial to their function [80,81,82]. Examples are also emerging to show that IDPs can remain unstructured, even in specific complexes and functional assemblies [83,84,85,86,87,88,89]. Figure 1D illustrates how the N-terminal transactivation domain of tumor suppressor p53 remains highly dynamic in the specific complex with cyclophilin D, a key regulator of the mitochondrial permeability transition pore (PTP) [90]. Such a dynamic mode of specific protein interactions seems much more prevalent than previously thought [91,92,93]. Arguably, the key to a quantitative and predictive understanding of IDPs and their dynamic interactions is the ability to accurately describe their dynamic conformational equilibria under relevant biological contexts. Such a capability is also critical for developing effective strategies for targeting IDPs in therapeutics, where they are considered a promising but difficult new class of drug targets [94,95,96]. For example, the disordered C-terminal region of protein tyrosine phosphatase 1B (PT1B), a key protein in breast cancers, can be targeted by a small natural product, trodusquemine [97]. The drug’s binding induces a shift in the dynamic conformational equilibrium of the C-terminal region of PT1B that allosterically disrupts HER2 signaling and inhibits tumorigenesis [98].

**Figure 1 molecules-28-04047-f001:**
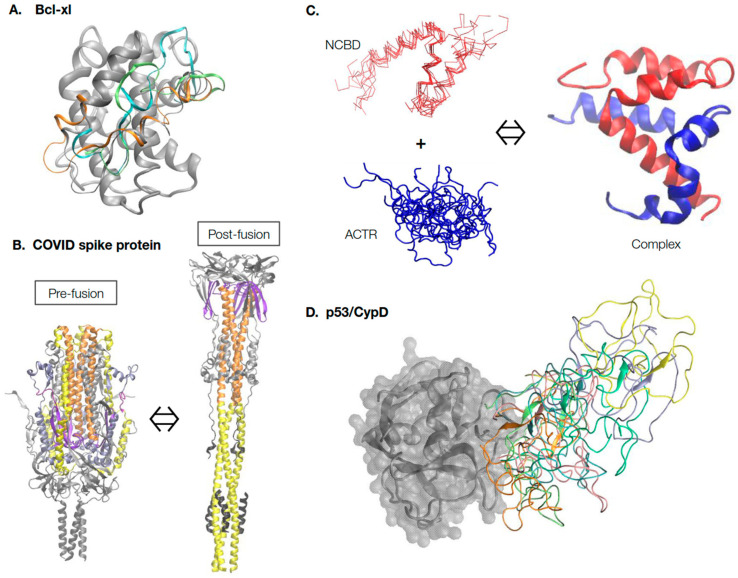
Continuum of protein structure and dynamics. (**A**) Inherent conformational dynamics of the BH3-only binding interface are crucial for the functioning of the Bcl-xl protein. Multiple representative conformations of the binding interface, shown in different colors, were generated using enhanced sampling simulations in explicit solvent [63]. (**B**) The COVID-19 spike protein undergoes dramatic large-scale conformational transitions in the pre-fusion and post-fusion states. The structures were extracted from Cryo-EM models (PDB: 6xr8 and 6xra [67]) and, for clarity, only common and resolved segments are shown. The central helices are shown in orange, heptad repeat 1 in yellow, and the fusion peptide proximal region in purple. Animations of the transition can be found on poteopedia.org. (**C**) Intrinsically disordered proteins ACTR and NCBD undergo binding-induced disorder-to-order transition to form the folded complex. The complex structure was taken from PDB: 1kbh [99], and the disordered ensembles of ACTR and NCBD were generated using coarse-grained (CG) MD simulations [100]. Note that while ACTR is fully disordered, free NCBD is a molten globule with essentially fully-formed helices. (**D**) Dynamic interactions of the N-terminal domain (NTD) of tumor suppressor p53 with the folded mitochondrial PTP regulator protein Cyclophilin D (CypD). CypD is shown in gray; multiple dynamic conformations of p53 NTD were extracted from previous CG MD simulations [90] and shown in different colors.

## 3. Generative Deep Learning for Biomolecular Modeling

Generative deep learning models are a class of neural networks that aim to learn the regularities of the training dataset and capture such regularities using appropriate probability distribution functions [54,101]. The learned probability distribution functions should be smooth, and can thus be used for generating new samples that are similar to the original dataset, with generally low computational costs. For biomolecular modeling, latent variable models such as variational autoencoders (VAEs) [102,103] and generative adversarial networks (GANs) [104] have been particularly suitable due to lower computational cost compared to sequential autoregressive models [105]. In both VAE and GAN models, the goal is to learn a lower-dimensional representation of the data in the so-called latent space, either explicitly (in VAEs) or implicitly (in GANs). VAEs learn a probabilistic encoder-decoder model that maps the input data to the latent space and back, while GANs train a generator that draws new samples from the latent space, and a discriminator that distinguishes between real and fake samples. The performance of both VAEs and GANs critically depends on careful training and tuning of the model hyperparameters, such that a good latent space representation exists and can be reliably learned.

Briefly, autoencoders (AE) are unsupervised ML algorithms that use neural network (NN) architecture to compress input data into knowledge (latent) representations. As illustrated in Figure 2A, an AE consists of two sub-models: an encoder and a decoder. The encoder is a generally a convolutional or feed-forward NN that reduces the high-dimensional input to lower-dimensional latent space, and the decoder NN transforms the latent space back to the original high-dimensional output. In other words, latent space is the input for the decoder. The number of encoder layers and decoder layers are generally the same. The AE is trained by optimizing the loss function to minimize the loss between the original and reconstructed outputs. For proteins, the input can be a set of Cartesian or internal coordinates. After training, the latent representation contains an information-dense and relatively lossless projection of the ensemble. Such latent spaces can be utilized in multiple ways. They could be used as CVs for calculating free energy surfaces, constructing Markov state models, or guiding additional MD simulations [106]. They could also be directly used for direct generation of new structures of the same biomolecule. For the latter, VAEs are often used where the distribution in the latent space is constrained to have a smooth normal distribution. GANs are also a type of unsupervised deep learning model; they have gained significant attention in recent years due to their ability to generate new data that is similar to a given training dataset. GANs also consist of two deep neural networks, namely a generator and a discriminator (Figure 2B). Generators are trained to generate new structures of the biomolecules by sampling from the prior distribution, whereas discriminators classify the generated structure as fake versus real. Both the sub-models are trained together in such a way that the generator produces structures that can fool the discriminator and discriminator is iteratively updated in order to accurately classify the structures.

## 4. ML Approaches to Identify CVs and Drive Enhanced Sampling in MD Simulations

A key application for ML in the generation of protein functional ensembles is to aid in the discovery of low-dimensional CVs that can distinguish important functional states, particularly those that could monotonically describe the transition between the states and are suitable for usage in enhanced sampling MD simulations such as umbrella sampling, adaptive sampling, and metadynamics [52,59]. Even though such approaches are not strictly generative models, a key advantage of integrating ML and MD is that the resulting conformations can be unbiased in order to yield proper thermodynamic ensembles. To be useful for this purpose, CVs must be of low enough dimension and, more critically, capture the slowest fluctuating degree(s) of freedom of protein conformation transitions [107]. The later requirement is highly nontrivial, in order to account for complex diffusive protein dynamics. Slow dynamics in other degrees of freedom not captured in the CVs leads to hidden barriers and hinders MD sampling [108]. It remains debatable if high-dimensional conformational fluctuation of biological proteins could be effectively reduced to representation using a few “reaction coordinates” (RCs) [109]. As such, data-driven discovery of CVs for protein dynamics remains an active area of research in recent years [110,111,112,113,114,115]. In the following, we highlight a few representative deep learning approaches in this direction.

In the so-called deeply enhanced sampling of protein (DESP) method developed by Salawu [110], a neural network (NN) is trained alongside the MD simulations to learn the latent space for representing the conformational space sampled. Biasing potentials are then introduced in the latent space, using the Kullback-Leibler (KL) divergence, to discourage MD from revisiting the conformational space already sampled. This approach was evaluated using a model helical peptide A_12_ and a small protein GB98. The results suggest that DESP can sample much broader conformational space compared to both conventional and accelerated MD within the same amount of simulation time. Importantly, unbiased probability distributions also could be uncovered. DESP is in principle generalizable to other proteins. However, additional work is required to understand how the quality of the latent space learned, as well as the choice of biasing potentials, affect the sampling efficiency. Further analysis is also required to evaluate how accurate and/or physical the additional conformational space sampled by DESP is. The apparent lack of reversible transitions in the DESP trajectories is concerning, which may reflect a critical limitation of using KL divergence directly to drive the sampling of new conformational space. Tao and co-workers [114] have compared the efficacies of AE and VAE for driving MD exploration of protein conformational space. Here, random points were selected in the latent space to initiate new MD trajectories. Using the adenosine kinase as a model system, it was shown that the latent space learned in VAE is superior to the AE-derived one for generating unsampled conformational space, likely due to the normal and smoother latent space distribution in VAE.

Tiwary and co-workers recently adapted reweighted auto-encoded variational Bayes for enhanced sampling (RAVE) for efficient sampling of protein loop conformations [112]. RAVE is based on the principle of the predictive information bottleneck (PIB), a predictive model for describing the evolution of a given dynamical system that encodes high dimensional input into low dimensional representations. PIB can be learned in an iterative manner, similar to autoencoders, and interpreted as the RC for usage in metadynamics enhanced sampling. The input to RAVE was generated using the automatic mutual information noise omission (AMINO) method to reduce the redundancies among a large set of raw order parameters (OP) that reflect generic features of protein contacts. The RC in RAVE is constructed as a linear combination of selected bias functions of the OP output from AMINO. Metadynamics trajectories generated using the RC as a bias variable are fed back to RAVE to further optimize the RC. This iterative process between enhanced sampling and RAVE learning continues until multiple transitions between different metastable states are sampled. Applied to protein T4 lysozyme, it was found that the functional states and free energy surfaces generated using the above protocol successfully recapitulate the loop conformational stabilities of the wild-type enzyme and three mutants. Furthermore, it was observed that the number of OPs required for the RC decreases in the mutants, suggesting increased cooperativity of conformational fluctuation (and thus decreased global flexibility). This work represents an interesting development towards an automated procedure for atomistic simulations of protein loop dynamics. It is not clear, though, whether the RAVE/AMINO approach can be generalized in order to sample either large-scale conformational transitions or disordered protein conformational ensembles (classes 2–4 of Figure 1).

Reinforced learning (RL)-based algorithms [116,117] have also been adapted for promoting the exploration of slow degrees of freedom of complex protein conformational fluctuations. Recently, a multiagent RL-based adaptive sampling (REAP) approach has been designed for sampling of rare states along user-defined CVs [113]. REAP is initialized by running short MD simulations, followed by conformational clustering to discretize the action space. The smallest clusters are selected as candidates. The reward is calculated for each cluster and optimization is performed by summing the reward as a weighted sum of the candidate CVs. Conformations with the highest rewards initiate the new set of simulations, minimizing redundant exploration. The process is repeated until either convergence or the desired final state is reached. REAP is effective if the user-defined CVs capture well the range of conformational space to be sampled. Multiagent LEAP allows more effective sampling by learning from independent simulations initiated from distinct starting conformations. This is achieved by introduction of a *stakes* function to modulate how rewards are attributed to different agents for discovering new states. The benefit of this multiagent formulation comes from two features. Conformations are labeled and utilized by the agent who discovers them, and therefore, each agent computes the rewards from different data points. Additionally, the agents share information within an action space to tell other agents what conformations they have already discovered. It was demonstrated that the multiagent REAP algorithm is more efficient in sampling the loop motion of Src kinase and driving large-scale conformational transitions of the transporter OsSWEET2b. The same multiagent formalism was shown to work well with other adaptive sampling techniques, including “least counts” and AdaptiveBandit. A key limitation of both single-agent and multi-agent REAP algorithms, though, is their dependence on user-defined CVs for prioritizing sampling. As illustrated in the works above, identification of such CVs can be highly nontrivial for protein conformational fluctuations in general. It is also not clear how one can reconstruct unbiased ensembles from REAP trajectories, a capacity which will be important for functional studies. The pros and cons of REAP and related adaptive sampling techniques remain to be thoroughly examined, especially compared to weighted ensemble methods that can recover both kinetic rates and thermodynamic stabilities of long-timescale processes [118].

## 5. Directly Sampling Conformational Space using ML-Derived Latent Representation

Learning low-dimensional representations of protein conformational ensembles is of interest for more than driving MD sampling. In principle, if one can learn the latent space representation from a relatively limit set of conformations, (e.g., generated by short MD simulations), one could then sample in the latent space to directly decode and generate new high-dimensional structures of the same protein. The requirement here is that the latent space representation is smooth and continuous. Such a generative approach can be dramatically faster than MD simulations. Nonetheless, the complexity and sheer size of the conformational space should never be under-appreciated, even for small proteins. Functionally relevant conformational states are vanishingly small in comparison, and importantly, they can be well-separated in the Cartesian space. It is not entirely clear if the desired mapping of high-dimensional protein conformational space to a smooth low-dimensional latent space representation exists, or, if such a mapping can be reliably learned from an incomplete set of pre-generated conformations. Nonetheless, direct sampling of protein conformational space using ML-derived latent representation is a highly attractive strategy and has continued to attract intense interest in recent years [119,120,121,122,123].

The ability of using the latent space to generate physically plausible structures has been evaluated using a set of proteins of different sizes, topologies and dynamic properties [119]. The results suggest that over 98% of reconstructed structures generated by AE for rigid proteins were classified as valid by a random forest (RF) classifier. For flexible proteins with multiple functional states, it was observed that VAE trained using both open and closed conformations could provide a reliable interpolation between these states. However, if only one of the states is available, VAE would fail to provide accurate extrapolation to distinct states not seen in the training set. This observation highlights the challenges of using latent presentation in direct sampling of novel conformational spaces of complex biomolecules. Hay and co-workers [121] tested whether an AE could be used to map MD generated conformations onto a pre-defined low-dimensional space (e.g., the first and second principal components) for subsequent prediction of new conformations, as a proof-of-principle. Using a flexible short peptide Ala_13_ and the protein calmodulin (CaM), the approach demonstrated modest success and could be most suitable for generating new initial structures for seeding additional MD simulations. Using pre-defined latent space is unusual and likely will be ineffective in general.

Degiacomi and co-workers [120] later described a 1D convolutional NN (CNN) that was directly trainable with protein structures to learn a latent representation of the conformational space. To address the challenge of extrapolating to novel conformational spaces, a key development here was the design of a new physics-based loss function that resembled the classical molecular mechanics force field. The loss function contained physics-motivated terms to enforce the covalent geometry and minimize steric clashes. Applied to a protein enzyme MurD with multiple open, close and intermediate states available, it was demonstrated that the CNN trained with physical constraints was capable of predicting the correct transition paths between the open and close states without any intermediate conformation provided. Intriguingly, the authors further showed that they could transfer features learned from one protein to others, which dramatically reduces the number of training samples for one protein and provides superior performance in generating novel low-energy conformations. It will be very interesting to see how this approach can be extended to generate dynamic protein ensembles in general, such as those of dynamic loops or IDPs.

Success has also been demonstrated recently using VAE to learn the dynamic conformational space of IDPs using short MD simulations and then generate full disordered ensembles [123]. Here, the objective was to use the smallest amount of MD sampling and generate the most accurate full ensembles, equivalent to those derived from much longer MD simulations. Three IDPs, namely, Q15, Aβ40, and ChiZ, were used as model systems where multi-μs MD trajectories had been previously generated. The dimension of the latent space was empirically chosen to be 0.75 *N*_res_ (number of residues). It was shown that only ~ the first 10% of the MD trajectories were necessary to regenerate the full ensemble of small IDP Q15, but the amount of training data required increased rapidly for larger IDPs such as ChiZ. The authors also evaluated the effects of latent space dimension and input features (e.g., dihedral angles instead of Cartesian coordinates). The results showed that larger latent vectors do not improve accuracy and the choice of input coordinates does not significantly impact model performance. Curiously, evaluation of VAE performance has focused on the accuracy of regenerating individual conformers, as measured by root-mean-square-distance (RMSD). It would be more desirable to examine ensemble distributions of key local and global properties, such as overall size, residual structures, long-range contacts, etc. One could also examine the details of conformational substates using clustering and principal component analysis, in comparison to the available ensembles generated by much longer MD trajectories. Furthermore, the three IDP ensembles appear largely random and devoid of nontrivial local structures and transient long-range organizations. Yet, many biologically relevant IDPs, especially those involved in cellular signaling and regulation, are not random coils, and contain important residual structures [124,125]. As such, it remains to be established whether such relatively standard VAE framework would be adequate for generative modeling of IDPs in general.

## 6. One-Shot Generation of Dynamic Protein Conformational Ensembles

An ultimate “one-shot” generative ML model would take the sequence of the protein and generate a full ensemble of the most relevant conformations. This is an extremely ambitious goal for proteins with high-dimensional and complex conformation space. Progress towards this goal has thus far been more limited [126,127,128]. Boltzmann generators are a type of generative model directly trained on the energy function of the system [127]. They can learn to sample from equilibrium distributions without directly learning the probability density function (e.g., from short MD trajectories). Instead, it learns to sample from a dimensionless energy function u(**x**) using a generative network and reweighting procedures. The generative network maps latent space samples from a simple prior distribution P(**z**) (e.g., Gaussian) to high-probability samples from the target distribution P(**x**)~e^−u(**x**)^. The probability of generating a configuration can be computed using the change-of-variables equation if the generative network is an invertible transformation. Invertible networks are called flows, and they can be stacked to create deep invertible neural networks. Even though the loss function can be designed to balance the sampling of low-energy states and the diversity of the sampled conformational space, Boltzmann generators show a strong tendency to suffer from mode collapse and generate similar conformations from a single metastable state. Instead, it needs to be combined with training by example, from existing experimental structures or short MD simulations. Additional terms can also be added to drive Boltzmann generators to sample along a pre-defined RC. This framework has mainly been demonstrated on relatively simple toy systems. Application to complex molecules such as proteins will lead to unrealistic structures with distorted covalent geometries and severe atomic overlaps. Instead, the technique requires careful separation of various degrees of freedom into Cartesian (backbone) and internal (sidechains) coordinate sets. It was demonstrated that the eventual generator could sample a key X-O loop conformational transition of protein BPTI that occurs on a millisecond-timescale. Clearly, extension of Boltzmann generators for one-shot generation of dynamic protein conformational ensembles requires much additional work. It has been argued that training solely on the energy is unlikely to be adequate [128].

Feig and co-workers recently described a conditional generative model for generating disordered protein conformational ensembles using the sequence alone as input [126]. The model, referred to as IdpGAN, is trained with long MD trajectories of a large set of IDPs, using a standard GAN architecture with multilayer perceptrons. The feasibility of IdpGAN is first demonstrated using trajectories from coarse-grained (CG) MD simulations. The results show that IdpGAN is highly effective in generating realistic disordered ensembles for an arbitrary IDP sequence, as characterized by a number of metrics, including protein overall dimension, contact distributions, and correlation of multiple pairs. The approach was further demonstrated by retraining the model using all-atom implicit solvent simulations. Impressively, conformational ensembles generated by IdpGAN do not merely capture overall structural properties such as compaction, but also contain realistic sequence-specific local structures such as residual helices. This is a highly nontrivial result that highlights the great potential of IdpGAN. It has been noted that IdpGAN relies on training on exhaustive trajectories of a carefully curated set of IDPs. This is a requirement that may be extremely difficult to meet for larger IDPs with nontrivial local structures and transient long-range interactions (such as proteins p53 [129] and tau [130]). It will be interesting to further evaluate the resolution of IdpGAN, for example, to resolve the effects of mutations of PTMs at one or few sites, which may be important in studies of regulatory IDPs.

## 7. Conclusions and Future Directions

Recent breakthroughs in ML approaches have transformed the studies of protein structure and function. The problem of predicting a native structure of protein has essentially been solved. A key new frontier is expanding these ML approaches to help describe dynamic fluctuations and transitions of proteins that are crucial to its functions. For this, much effort has been dedicated to generative ML models such as VAEs and GANs (Figure 2). A common objective of these generative models is to learn a low-dimensional latent representation of the high-dimensional conformational space of the protein, which can then be used to guide additional MD sampling and construct free energy surfaces. If restrained to be smooth and continuous, the latent space could also be used for direct generation of novel conformations not seen in the training sample. A fundamental challenge, however, is that the possible conformational space of proteins is vast, and functionally relevant states are vanishingly small and often well segregated. It remains to be established to what degree the dimensionality of protein conformational space could be reduced, whether the complex and discontinuous distributions of low-energy states could be mapped to smooth and continuous ones in the latent space, and how the mapping can be reliably learned. As such, only limited successes have been demonstrated at this point. Generative modeling of dynamic protein conformations has only been feasible in relatively simple cases such as protein loop motions, or when sufficient examples are available for training, such as for relatively simple transitions between to known states or for highly disordered proteins with limited nontrivial local and global structural features.

It is generally recognized that it is likely critical to incorporate physics towards generative ML models that may be more generally applicable to nontrivial proteins [106,131,132,133,134]. For example, Boltzmann generators are directly trained on the energy functions of the system to generate independent samples of low-energy states, even though it is also evident that training on energy alone is unlikely to be adequate for complex biomolecules [127,128]. It should be noted that the physical principles of molecular interactions arguably contain all the information needed for generating the complete conformational ensemble of any protein, though this method is generally impractical due to its computational cost. On the other hand, the large number of available experimental structures, as well as numerous high-quality MD trajectories on many proteins, may already contain enough of the information needed to train transferable ML models for generating dynamic protein conformational ensembles, given the sequence. Developing such models will require advances on multiple fronts.

Deeper and more systematic understanding of if and how the complexity of the high-dimensional conformational space of proteins can be mapped to a smooth and continuous space of sufficiently low dimensionality. This may be investigated using ultra-long trajectories of a large set of folded and unfolded proteins, such as those from Shaw and co-workers, using the latest balanced atomistic protein force field [49], as well as those generated using extensive enhanced sampling simulations [124,125,135]. A challenge here is to ensure the quality and convergence of the structural data, particularly for IDPs.Consistent and rigorous evaluation of the performance of generative ML models that is widely accepted and adopted by the community. At present, most developments are benchmarked using different custom examples and special test-cases. There is only minimal cross-comparison needed to rigorously establish the strengths and weaknesses of various approaches, such as over-fitting, computational cost, and interpretability. Again, ultra-long MD trajectories or highly-converged conformational ensembles generated from enhanced sampling simulations of proteins of different sizes, topologies and dynamic properties could be used as a standard benchmark set. Similar practices have been instrumental in the development of methods for predicting protein structure [8], protein-protein interaction [136], and protein-ligand interaction [137].New ML approaches for integrating information from diverse datasets, such as protein structures, sequence alignment, and MD trajectories, and incorporating physical principles of molecular interactions, such as various empirical protein energy functions. ML has been under accelerated development in recent years, with many exciting ideas emerging, such as large language models (LLMs) [12], deep transfer learning [138], and diffusion models [139]. Furthermore, the computational infrastructure now allows much larger and increasingly complex models to be trained with extremely large datasets [140].

At present, it is hard to clearly envision how generative ML models of protein conformational ensemble may look or how generally applicable these models will be across the spectrum of protein dynamics relevant to biological function (Figure 1). However, development of ML and artificial intelligence tools has rapidly accelerated in recent years, and the trend should continue in the foreseeable future. Notwithstanding the challenges described in this review, generative modeling of protein conformational dynamics has immense potential to completely transform how we study protein structure, dynamics, and function in biology and medicine, and therefore offers a bright future for the field.

## Figures and Tables

**Figure 2 molecules-28-04047-f002:**
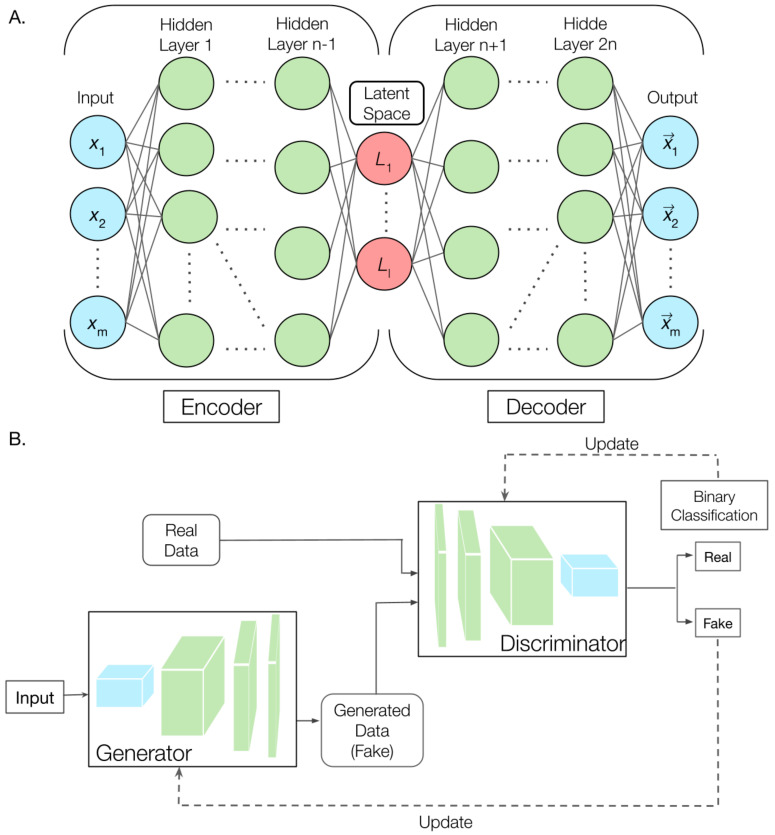
Generative ML approaches for protein dynamics. (**A**) An autoencoder consists of two fully-connected NN. The encoder compresses the *m*-dimensional input data vector (x_1_, x_2_, …, x_m_) in several orders in dimensions into the *l*-dimensional latent space vector (*L*_1_, *L*_2_, …, *L*_l_, *l* << *m*), while the encoder decompresses the latent space and reconstructs the original high-dimensional form ($\vec{x_{1}},\vec{x_{2}},\ldots ,\vec{x_{m}}$). (**B**) GANs contain two networks. The generator takes input that is of a predefined latent space and generates a similar output to realistic data using a deconvolutional neural network. The discriminator is a convolutional neural network that classifies whether the data is real or fake. The generator is further updated by using this discriminator output to perform better in the next epoch in order to fool the discriminator, and the discriminator also backpropagates to update and reduce loss in order to more accurately classify the data.

## Data Availability

Not applicable.

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
