# Peer review of "Machine Learning Generation of Dynamic Protein Conformational Ensembles"

_molecules, 2023, doi:10.3390/molecules28104047_

Round 1

Reviewer 1 Report

This paper reviews machine learning techniques for generating protein conformation. I did not see any issues with the machine learning part of my specialty, except for the following. I think this review is very interesting and important, also well-written. I would like to leave the more detailed parts to more advanced experts.

1) The caption of Figure 2.  (x,y,z) is easily mistaken for a tuple. It should be made clear that the parentheses have no mathematical meaning such as (e.g., x,y,z). Also, the definition of m is not clear, though it may be the number of samples or the input dimension.

Author Response

Reviewer Comment: 1) The caption of Figure 2.  (x,y,z) is easily mistaken for a tuple. It should be made clear that the parentheses have no mathematical meaning such as (e.g., x,y,z). Also, the definition of m is not clear, though it may be the number of samples or the input dimension.

Author response: thank you very much for the positive comments. We have updated the Figure 2 caption to clarify that m (and l) refers to the dimensions on the input data (and latent space vector). We also include the wording "vector" in front of ( ... ) to clarify that the parenthesis denotes data vectors.

Reviewer 2 Report

Machine Learning Generation of Dynamic Protein Conformational Ensembles

The authors present a review of machine learning developments in the field of protein conformational ensembles, focused on the most recent developments. The manuscript is well-written, but would benefit from a more consistent discussion of the three key themes mentioned in the Abstract, including in the Conclusion more of an assessment of progress against these three themes.

Major comments:

Whilst this is not a systematic review, it might be useful to include a brief methodological explanation of which databases were searched and with what keywords, and what inclusion or exclusion criteria were employed in the review. Even if the authors primarily used their own knowledge of recent publications, all reviews should include some sort of comprehensive database search. This is not a required change, but something the authors may wish to consider.

The Article would also benefit greatly from consistency in its three key themes. The Abstract states that the "problems can be grouped into three increasingly challenging classes". The article should then follow this Abstract and reference these challenges as much as possible, and the Conclusion should also reference these three challenges. If the Abstract does not match the Discussion and the Conclusion in terms of the key points raised, it is harder for the reader to follow the thoughts and conclusions of the authors. Specifically in the Conclusion, the reader's attention is focused on three issues (e.g. diverse datasets), but these are not the three issues from the Abstract.

In addition to the above point, the article would benefit from numbered headings. After Introduction, it would be helpful to have Discussion, with following headings being subheadings of Discussion, e.g. X.0 Discussion and X.1 A rich continuum of protein structure and dynamics for function (or whatever revised headings are used).

Minor comments:

There are several small typographical errors, as is usually the case for any manuscript. Some are highlighted below.

Page 3, first line: "recent works published in the three years", the sentence is missing the word 'last' or similar

Page 8, "bayes" should be Bayes (it is incorrectly capitalised in the referenced article as well)

Page 10, last paragraph, "Success has also been demonstrated recently for using VAE to ..." delete the word 'for'

Page 13, 3rd point, last sentence, "with on extremely large datasets" the word 'on' should be removed

Page 13, writing the last words of a paper is always difficult, suggest to replace:

"The future can thus be expected to be extremely bright for generative modeling of protein conformational dynamics. These models could completely transform how we study protein structure, dynamics and function in biology and medicine in the not too distant future."

With the more compact:

"Notwithstanding the challenges described in this work, generative modeling of protein conformational dynamics has the potential to completely transform how we study protein structure, dynamics and function in biology and medicine, and so offers a bright future."

Aside from some minor typographical errors or small instances of missing / extraneous words, the manuscript is well-written.

Author Response

Major comments:

Whilst this is not a systematic review, it might be useful to include a brief methodological explanation of which databases were searched and with what keywords, and what inclusion or exclusion criteria were employed in the review. Even if the authors primarily used their own knowledge of recent publications, all reviews should include some sort of comprehensive database search. This is not a required change, but something the authors may wish to consider.

Author Response: We appreciate the valuable suggestions and have now included a statement on how the works included in this review were identified. As stated in the last paragraph, we focus on “recent progresses in development and application of deep generative models for biomolecular modeling, particularly for generating dynamic structure ensembles of proteins in various contexts.”

The Article would also benefit greatly from consistency in its three key themes. The Abstract states that the "problems can be grouped into three increasingly challenging classes". The article should then follow this Abstract and reference these challenges as much as possible, and the Conclusion should also reference these three challenges. If the Abstract does not match the Discussion and the Conclusion in terms of the key points raised, it is harder for the reader to follow the thoughts and conclusions of the authors. Specifically in the Conclusion, the reader's attention is focused on three issues (e.g. diverse datasets), but these are not the three issues from the Abstract.

Author Response: We agree with the critique on the abstract could read a bit misleading and have now extensively revised the abstract to be more consistent with the focus and organization of the review.

In addition to the above point, the article would benefit from numbered headings. After Introduction, it would be helpful to have Discussion, with following headings being subheadings of Discussion, e.g. X.0 Discussion and X.1 A rich continuum of protein structure and dynamics for function (or whatever revised headings are used).

Author Response: We agree with the suggestion and have now numbered all headings.

Minor comments:

There are several small typographical errors, as is usually the case for any manuscript. Some are highlighted below.

Page 3, first line: "recent works published in the three years", the sentence is missing the word 'last' or similar

Page 8, "bayes" should be Bayes (it is incorrectly capitalised in the referenced article as well)

Page 10, last paragraph, "Success has also been demonstrated recently for using VAE to ..." delete the word 'for'

Page 13, 3rd point, last sentence, "with on extremely large datasets" the word 'on' should be removed

Author Response: Thank you for the careful proofreading. These errors have been corrected.

Page 13, writing the last words of a paper is always difficult, suggest to replace:

"The future can thus be expected to be extremely bright for generative modeling of protein conformational dynamics. These models could completely transform how we study protein structure, dynamics and function in biology and medicine in the not too distant future."

With the more compact:

"Notwithstanding the challenges described in this work, generative modeling of protein conformational dynamics has the potential to completely transform how we study protein structure, dynamics and function in biology and medicine, and so offers a bright future."

Author Response: we greatly appreciate the wonderful suggestion and have revised the closing sentence in the revised manuscript: “Notwithstanding the challenges described in this review, generative modeling of protein conformational dynamics has immense potential to completely transform how we study protein structure, dynamics and function in biology and medicine, and therefore, offers a bright future for the field.”

Reviewer 3 Report

The research proposal provides a comprehensive overview of the current state of machine learning applications in predicting protein dynamics, focusing on the different functional classes of protein dynamics and the challenges associated with each class. The proposal also delves into generative deep learning models, specifically Variational Autoencoders and Generative Adversarial Networks, and their applications in biomolecular modeling. Overall, this is a very interesting review.

I have a few comments: 

  1. The proposal should address the risk of overfitting in the machine learning models, particularly given the complexity of protein structures and dynamics. The authors should discuss how they will ensure that their models generalize well to new, unseen protein data.

  2. The interpretability of the machine learning models is crucial, especially in biological applications. The proposal should discuss how the researchers plan to interpret the models' predictions and ensure that their models provide biologically meaningful insights.

  3. The proposal should address the challenges related to obtaining high-quality and diverse structural data for proteins, particularly for intrinsically disordered proteins (IDPs). The authors should critically assess the limitations of current structural data and discuss strategies to overcome these challenges in their study.

  4. The authors should address the computational cost of their proposed machine learning models, especially in comparison with traditional molecular dynamics simulations. This should include a discussion of the trade-offs between accuracy and computational efficiency, and how these factors will be considered in the development and optimization of their models.

Author Response

We greatly appreciate the reviewer's support recommendation. We have now added a couple sentences to incorporate the important issues pointed out by the reviewer in the "Conclusions and Further Directions" section.

Round 2

Reviewer 2 Report

The authors have responded to all my comments satisfactorily, thank you.